# Sustainable Mechanism of the Entrusted Transportation Management Mode on High-Speed Rail and the Impact of COVID-19: A Case Study of the Beijing–Shanghai High-Speed Rail

**Chao Ji** [1], **Yanke Yao** [2,*], **Jianqiang Duan** [1,*] **and Wenxing Li** [1]

1   School of Economics and Management, Beijing Jiaotong University, Beijing 100044, China; 15113107@bjtu.edu.cn (C.J.); wxli@bjtu.edu.cn (W.L.)
2   School of Traffic and Transportation, Beijing Jiaotong University, Beijing 100044, China
\*   Correspondence: 20120941@bjtu.edu.cn (Y.Y.); jqduan@bjtu.edu.cn (J.D.)

**Abstract:** The transport management mode fundamentally determines the sustainable development of high-speed rail passenger transport (HSRPT), which was shocked by the COVID-19 pandemic in 2020. In order to study the sustainable development mechanism of HSRPT and the impact of COVID-19, primarily based on the data from the Beijing–Shanghai high-speed rail (HSR) taken from 2018, we adopt system dynamics (SD) to provide a scenario simulation method to examination sustainable operation status of HSRPT under the entrusted transportation management mode (ETMM) by VENSIM, and take into account the following two evaluation indicators: economic and operational. The results show the following: (1) Transportation demand and commissioned transportation management fees play a vital role in the sustainable operation of the Beijing–Shanghai HSR, causing significant changes in transportation revenue and transportation costs. (2) COVID-19 had a great impact on the sustainable operation of the Beijing–Shanghai HSR. In 2020, the turnover and transportation profit of the Beijing–Shanghai HSR fell by 74.31% and 49.19%, respectively. In 2022, the transportation profit can be restored to the level of 2019. The study results reveal that Beijing–Shanghai HSR under the ETMM has a good sustainable development capability.

**Keywords:** high-speed rail passenger transport; sustainable operation; entrusted transportation management mode; COVID-19 impact; system dynamics

## 1. Introduction

High-speed railway (HSR) is the important support and guarantee for national economic and social development. In recent years, Chinese HSR has developed vigorously. By the end of 2020, the operating mile of HSR has reached 38,000 km, ranking first in the world. The Beijing–Shanghai high-speed railway is a representative of the rapid development of Chinese HSR, whose operation and management mode adopts the entrusted transportation management mode (ETMM). Since the opening of the Beijing–Shanghai HSR in June 2011, it has operated safely for 10 years, the passenger volume and transportation revenue have increased year by year, and it has achieved good economic benefits, which indicates that the HSR under ETMM has good sustainable development ability.

Chinese railways generally adopt the integrated operation mode of network and transportation. HSR operating enterprises are not only responsible for the construction and maintenance of railway lines, EMUS, stations and other infrastructure, but also for daily railway transport production management, such as transport organization management and transport safety management. In this mode, HSR operating enterprises need to invest lots of manpower, time and capital costs. The proposal of ETMM is helpful to solve this problem. Under this mode, the HSR operating enterprises are firstly responsible for the

construction of HSR. After the completion of HSR, HSR operating enterprises sign the transportation entrustment agreements with railway bureaus along the HSR line, entrusting railway transportation, facility maintenance and other works to the railway bureaus, and HSR operating enterprises are responsible for formulating the passenger transport development strategy of HSR, supervising the process of high-speed rail passenger transport (HSRPT) and paying the entrusted transportation management fees. ETMM defines the responsibilities of entrusting party and entrusted party according to the transportation entrustment agreement, so that the two parties can have clear division of labor and cooperate with each other, which can meet the industry requirement of unified dispatch and command of railway network and also help to achieve the operation goal of maximizing asset efficiency.

The Beijing–Shanghai HSR adopts ETMM. The Beijing–Shanghai HSR Co., Ltd. completed the Beijing–Shanghai HSR construction work from 2008 to 2011. In 2011, Beijing Rail Bureau Group, Jinan Rail Bureau Group and Shanghai Rail Bureau Group were entrusted to carry out specific transportation and management of the Beijing–Shanghai HSR [1]. These entrusted railway bureaus are specifically responsible for transportation organization and management, transportation facilities and equipment management and maintenance, transportation safety production management, overhaul project implementation, statistical management and other contents.

Under the ETMM, the division of labor between Beijing–Shanghai HSR Co., Ltd. and entrusted Railway Bureau is clear. Beijing–Shanghai HSR Co., Ltd. proposed passenger product design and train diagram scheme; Entrusted Railway bureaus are responsible for train preparation, ticket sale, command and dispatch, passenger waiting, boarding and landing organization, passenger service, ticket checking and departure and other passenger transport links; finally, the Beijing–Shanghai HSR Co., Ltd. and the entrusted Railway Bureau jointly conduct quality assessment and information feedback on passenger transport operation, and put forward quality improvement opinions.

The Beijing–Shanghai HSR has been operating effectively after adopting the ETMM. On 6 January 2020, the Beijing–Shanghai HSR Co., Ltd. was listed on the main board of the Shanghai Stock Exchange, and once rose by the limit after the opening. The Beijing–Shanghai HSR has transported a total of 1.35 billion passengers from June 2011 to June 2021. It is the first HSR in China to achieve positive revenue.

In order to demonstrate that HSR under ETMM is sustainable both under normal conditions and under the impact of COVID-19, we take Beijing–Shanghai HSR as an example; the paper studies the sustainable operation mechanism of HSR passenger transport under the ETMM and the impact of COVID-19 on the sustainable development of HSR passenger transport. It mainly includes the following contents: (1) Expounding the definition of ETMM. (2) Analyzing the factors affecting the sustainable operation of HSR passenger transport under the ETMM. (3) Constructing the sustainable operation mechanism model of ETMM based on system dynamics (SD). (4) Taking the Beijing–Shanghai HSR as an example, set the index of sustainable development, use the SD method simulate the sustainable operation state of Beijing–Shanghai HSR in the next few years, change the values of passenger demand and unit entrusted transport management fees, analyze the impact of its change on the sustainable operation of the Beijing–Shanghai HSR, and verify the feasibility of the model. (5) Taking scenarios with different severity of COVID-19 as examples, analyze the impact of COVID-19 on sustainable development of Beijing–Shanghai HSR by comparing the simulated index change.

The innovations of this paper include the following: (1) We give the definition of the ETMM, put forward the influencing factors of the sustainable development of HSR passenger transport under the ETMM, and analyze the correlation among the influencing factors. (2) We use the SD method to establish the sustainable development model of HSR under the ETMM, and then we analyze the sustainable development mechanism of HSR passenger transport under the ETMM through the simulation of the Beijing–Shanghai HSR. (3) We introduce epidemic impact factors and predict the quantitative impact of different

epidemic impact degrees on the sustainable operation index Beijing–Shanghai HSR based on SD simulation.

The research contents of this paper are as follows: the second part introduces the relevant research literature. The third part establishes the sustainable operation model of ETMM based on SD. The fourth part takes the Beijing–Shanghai HSR as an example to carry out simulation and sensitivity analysis. The fifth part studies the impact of epidemic severity on the sustainable development of the Beijing–Shanghai HSR. The sixth part is the conclusion and prospect.

## 2. Literature Review

Scholars around the world have conducted various studies on the sustainable development of HSR, mainly including the sustainable development strategy of HSR, the operation and management mode of HSR, and the sustainability of the operation and management mode of HSR. Among them are as follows:

In terms of the sustainable development strategy of HSR, Li et al. [2] designed a sustainability assessment framework for HSR based on the following three aspects: operational efficiency, financial solvency and long-term capital, and proposed that strong passenger flow and affordability of ticket price are the dual guarantee for sustainable profitability of HSR. Ping et al. [3] put forward the need to further combine the latest progress of modern science with the construction, operation, maintenance and management of HSR, and support the sustainable development of China's HSR through innovation. Li et al. [4] proposed to grasp the prospective scale and construction pace of China's HSR and develop HSR by considering the needs of mass travel modes. Russo et al. [5] took the upgrading of HSR lines in Italy as an example, pointing out that in the planning and construction of HSR network, national strategic planning should be considered, and sustainable development should be pursued. They proposed HSR line construction schemes from lean, agile, flexible and green directions. Xue et al. [6] proposed that in the planning and construction of HSR, the degree of transparency and democratization should be improved to promote the coordinated development of HSR, society and environment. Rungskunroch et al. [7] pointed out that HSR services should cooperate with the city's policymakers, business groups and related experts, who can enable the full integration of HSR services with urban planning, societal values, and other externalities.

In terms of the operation and management mode of HSR, the existing studies mainly focus on the separation mode and integration mode of network transportation. HSR operation in Germany adopts the mode of separation of network transportation. Deutsche Bahn AG has several management departments to manage network operation, long-distance passenger transport, regional passenger transport and freight business, respectively, which brings revenue growth to Deutsche Bahn [8]. Pittman et al. [9] studied the integration of railway network carrying mode and network separation mode of competition issues, think network separation mode, take the network access to a third party payment open is an effective mode of introducing competition, and stresses the network after the separation of network and operating enterprises accounting separation, the importance of transparency to promote competition. Friebel et al. [10] verified that the separation mode of network and operation has the advantage of reducing operation cost for railway operation enterprises. Kurosaki et al. [11] pointed out the difference between the railway network separation mode in Britain and other European countries. Each transport company and road network company are independent from each other and do not interfere with each other in operation, calling it the "complete separation mode". Chinese railways basically adopt the integrated mode of network transportation. Chang et al. [12] analyzed that the mode of "separation between network and transportation" encourages competition and yields more profits, resulting in rich financial performance. Wu et al. [13] analyzed the reform direction of China's railway transport enterprises under the integrated mode of network transportation from the following three aspects: the responsibility of the subject of property rights, the adaptability of railway management system and market economy, and

the relationship between the system and enterprises. Xu [14] pointed out that in order to cope with the competition in the transportation market, China's HSRPT can continue to maintain the integrated network and transportation structure of HSRPT and implement a simple separation mode of network transportation in some lines.

In terms of the sustainable development of high-speed rail operation and management models, there are few relevant research documents. Among them, Gui et al. [15] proposed that for the sustainable development model of large-scale projects, it is necessary to consider the market work in operation, taking the HSR as an example to propose the necessity of China's railway market reform. Qin et al. [16] introduced the concept of revenue management in HSR operations and propose a mixed-integer nonlinear programming (MINLP) model, which appropriately captures passengers' choice behavior, to optimize the price and seat allocation for HSR simultaneously. In order to provide a flexibility pricing strategy for the sustainable development of the HSR, Qin et al. [17] divided the passenger market according to the different factors affecting passengers' choice behavior, maximized ticket sales revenue with expected travel cost as the reference point, and used prospect theory to construct a differentiated pricing model under elastic demand. Based on the theory of sustainable development, Yan et al. [18] constructed a quantitative evaluation model, a balanced development coefficient and an evolution analysis model of obstacles, and analyzed the development status and evolution laws of China's railway industry. Zhou [19] discussed that the future operation and management direction of HSR should promote the reform of the division of labor between government and enterprises to achieve the balance between economic and social benefits of HSR operations by studying the impact of HSR on industries in surrounding cities. Tian et al. [20] proposed that the government should oversee possible market failures in HSR development and facilitate regional economic development on the principle of ensuring both efficiency and fairness.

There is almost no research by foreign scholars on the sustainable development of ETMM. Chinese scholars mostly conduct macro qualitative analysis from the aspects of policies, modes and mechanisms, and lack quantitative analysis of sustainable development under ETMM. Hu et al. [21] studied the current situation of operation cost control under the ETMM of HSR and proposed the idea of improving the operation cost control mode. Song et al. [22] put forward countermeasures for the development of ETMM of Beijing–Shanghai HSR from aspects of strengthening communication and coordination of entrusted transportation management and creating market competition mechanism of ETMM, so as to promote the sustainable development of Beijing–Shanghai HSR. Zheng et al. [23] analyzed the connotation of entrusted transportation management of joint-venture railways and proposed suggestions and countermeasures for strengthening entrusted transportation management of joint-venture railways, in view of the problems existing in entrusted transportation management.

The outbreak of COVID-19 in 2020 has had a great impact on the sustainable operation of HSR, and it has made the sustainable operation of HSR a research hotspot. Wang et al. [24] analyzed the impact of the Wuhan city closure due to the epidemic on the Chinese railway system. The results show that the city closure of Wuhan has a small impact on the overall connectivity of China's railway network and has a greater impact on the railway system in regions such as the Yangtze River Delta; in the long run, it has little impact on the sustainable development of railway system. Park et al. [25] analyzed South Korea's traffic data before and after COVID-19 and proposed a plan to improve service convenience through untact technology, establish a safety prevention system, and improve the reliability of railway transportation. Lin et al. [26] analyzed the impact of the new crown epidemic on the development of passenger and freight transport and infrastructure investment in China's railway industry, and proposed the following countermeasures: establishing a long-term mechanism for preventing and controlling public emergencies, and accelerating the construction of digital and intelligent railways. Jiao et al. [27], focusing on the sustainability of the high-speed rail network, proposed a weighted network efficiency metric to assess network performance and calculate the influence of lockdowns during the COVID-19

outbreak in overall network efficiency. Wang et al. [28] analyzed the daily railway passenger volume data during the Spring Festival travel rush and established a RegARMA model to predict the COVID-19 influence on passenger flow of HSR. Wang et al. [29] analyzed the changes in passenger and freight volumes, fixed asset investment levels, and transportation costs in the whole society since the outbreak of the new crown epidemic. Transportation costs have increased greatly. Werner [30] and Ku [31] pointed that the COVID-19 pandemic has had an impact on long-distance railway passenger transport in Asia and Europe and supported that the government should use stimulus funding packages for improving the attractiveness of public transportation and ensuring the sustainability of railway passenger transport. Based on this, Alessio et al. [32] pointed that railway sector should undergo the steps named the five "R"s (resilience, return, reimagination, reform, and research) to better continue providing services throughout future crises.

In terms of research methods, many scholars around the world use the system dynamics (SD) method to study the sustainable development of systems. SD can be used to predict the economic benefits generated in the future [33–36], as well as analyze the key factors influencing the system [37–39]. Zuo et al. [40] established a sustainable development model called the 3E system (economic-energy-environment System) by using the SD. The results showed that the long-term development of the 3E system in the Beijing–Tianjin–Hebei region was not sustainable, but it can be changed by adjusting the energy structure and increasing the investment in environmental protection, so as to improve the environmental quality. Liao et al. [41] took Jiuzhaigou Scenic Spot in China as an example and constructed an SD model based on the analysis of economic and environmental subsystems to promote sustainable development. Xue et al. [42] used the SD theory to construct the transit metropolis SD simulation model from the four subsystems of economy, society, environment, and transportation supply and demand to predict the quantitative indexes of transit metropolis construction.

In using system dynamics to analyze aspects of the sustainable development of railway operation, Xue [43] used the SD approach to quantitatively analyze the cumulative effects of different private capital investment models in public transport from the environmental perspective. Li [44] comprehensively adopted SD and game theory methods to simulate the sustainable development and evolution mechanism of high-speed rail operations, and proposed safety management and control recommendations for HSR operations under the ETMM. Stefan [45] examines the nexus between the main forms of railway transport, related investments, specific air pollutants, and sustainable economic growth, a bidirectional causal relation was noticed between the length of the railway lines, investments in railway transport infrastructure, and GDP. Jin et al. [46] used an SD model to analyze the relationship between the sustainable development of HSR and the choice of passenger demand patterns and the planning of train stops. Miao [47] applied an SD method and logistic model to simulate the growth of intercity HSR passenger flow and its share rate, and put forward sustainable development countermeasures. Mou et al. [48] established an SD model of HSRPT demand to analyze the relationship between economy, population, travel factors and passenger flow growth, and predict the sustainable development and operation of high-speed rail in the next few years.

At present, there are few studies using SD simulation method to analyze the impact of COVID-19 on the passenger volume of HSR, and there are almost no papers on the sustainable development of HSRPT with ETMM. Taking the Beijing–Shanghai HSR as an example, this paper analyzes the sustainable development mechanism of ETMM of HSR by using an SD method, and it simulates and predicts the quantitative impact of COVID-19 on sustainable operation of HSR.

## 3. Methodology

System dynamics (SD) is a theory that uses the system structure, causal loop of each link and feedback loop to establish a comprehensive model and solve the system performance through simulation. System dynamics method can be used to analyze the

relationship between the influencing factors of high-speed rail passenger transport (HSRPT) sustainable operation under ETMM, analyze its sustainable operation mechanism and deduce the future development tendency.

*3.1. Causal Loop Diagrams (CLD) Analysis*

Under the ETMM, analyzing the influencing factors in the HSRPT operation system and the logical relationship between the factors is the basis of establishing the SD model.

3.1.1. Analysis of Influencing Factors

In this paper, we select operation indicators and economic indicators to represent the sustainable development of HSRPT under the ETMM. Among them, passenger turnover is selected as operation index, total transportation revenue, transportation profit and operating costs are selected as economic indexes.

Focusing on the indicators of sustainable development, the factors influencing the sustainable development of HSRPT under the ETMM mainly include the following: operating cost, market competition, transportation supply, transportation demand, and transportation price.

(1) Operating cost

Operating cost refers to the economic input in the operation of HSRPT. Unlike the general operation mode, under the ETMM, the transport organization management, transport facilities and equipment management, transport safety and production management of HSRPT are all entrusted to the railroad bureau group along the railroad line. The ETMM focuses on providing entrusted transportation management expenses.

Operating costs are divided into annual operating costs and unit operating costs. Annual operating costs are mainly composed of ETM fees, EMU usage fees, road network usage fees, power supply and electricity maintenance, energy costs, fixed costs, R&D costs and other costs [1]. ETM fees include EMU service fees, equipment maintenance and station passenger service fees. Unit operating cost refers to the operating cost per unit of passenger turnover, which is the ratio of annual operating cost to annual passenger turnover.

(2) Market competition

Market competition refers to the competition between passenger transport products in different modes of transport. Various modes of transportation have different technical and economic characteristics, and when there are two or more modes of transportation in the same region, there is competition between them.

For travel distances of about 100 km, the travel time of HSR is similar to that of highway, but less convenient than highway. For travel distances of about 300–500 km, the travel time of HSR is close to that of airlines, but considering the comfort, punctuality and convenience of HSR, it has a higher competitiveness compared to highways and airlines. For travel distances of 800–1300 km, airlines and HSR have relatively fierce competition. For travel distances above 1300 km, airlines have a more obvious advantage over HSR.

Considering market competition among passenger transportation products, the focus is on the fare comparison relationship between different transportation modes, and determining the relationship between HSR fares, civil aviation fares, and highway fares according to the comfort and convenience of different transportation products before.

(3) Transportation supply

Transportation supply refers to the passenger services and passenger products that can be provided by HSRPT operators, mainly including the supply capacity of HSR and the convenience of passenger transport products. The supply capacity of HSR refers to the passenger turnover that HSR operating companies can undertake, which is mainly expressed in terms of the number of trainsets that HSR passenger operating companies can provide. The degree of convenience of passenger transport products is mainly expressed by the frequency of trips. The higher the frequency of trips, the higher the degree of convenience, vice versa, the convenience will decrease.

(4) Transportation demand

Transportation demand is the need of passengers in terms of spatial displacement. Transportation demand with the desire to travel and the ability to pay at the same time is a prerequisite for passengers to purchase transportation products. Transportation demand is a vital factor in achieving sustainable operation of HSR passenger operating companies.

The demand for HSRPT is the sum of the passenger transport miles of high-speed railway (HSR) required by passengers due to displacement demand. Passenger demand for HSRPT is closely related to regional economic conditions, population conditions, regional passenger group characteristics and other factors. At the same time, for HSRPT operating companies, the ability of residents to pay for travel needs to be considered when providing passenger transport products. The ability of residents to pay for travel is measured in relation to the living standards of urban and rural residents and their population share in a certain region. According to the analysis method of residents' living standard, the national residents' ability to pay for travel in 2019 is estimated to be 0.76 yuan/person-kilometer [49].

(5) Transportation price

Transportation price is the fee charged to passengers for the completion of the passenger service by providing passenger transportation products. The setting of transport prices plays a crucial role in the operation of high-speed passenger transport, which interacts with operating costs, market competition and transport demand factors, and also directly determines transport revenues and transport profits.

In this paper, two quantitative price indicators are given, which are target tariff and set tariff. Target tariff refers to transport prices based on cost-led pricing, established on the basis of unit operating costs, while set tariff refers to demand-led pricing, taking into account competitive market factors, related to the ability of residents to pay for travel and to civil aviation fares and highway fares.

### 3.1.2. Causal Loop Diagrams CLD

In HSRPT, as passenger turnover changes, operating costs change, along with the target tariff. Changes in passenger turnover and target tariff cause changes in transport profits, which bring changes in railroad supply, and supply and demand determine railroad passenger turnover, and so on, forming a dynamic feedback structure [50].

Similarly, the set price, determined by competition or the ability of the population to pay for travel, has the same role in the feedback structure, where the setting of the tariff determines the transport profit as well as the transport revenue, but also the passenger turnover, which determines the economic efficiency as well as the operational indicators of the high-speed passenger transport and affects its sustainability. The casual loop diagram among the factors influencing the sustainable development of HSRPT under the ETMM is shown in Figure 1 below. The arrows indicate the logical cause–effect relationships among the factors, while "+" and "−" indicate positive and negative effects, respectively. The causal feedback loops shown in Figure 1.

(1) "Passenger turnover → + Total transportation revenue → + Transportation profit → + the number of EMUs → + HSR supply capacity → + Passenger turnover"

(2) "Passenger turnover → + Annual operating costs → − Transportation profit → + the number of EMUs → + Frequency of trains → + HSRPT demand → + Passenger turnover"

(3) "Passenger turnover → − Unit operating cost → + Target tariff → + Total transportation revenue → + Transportation profit → + the number of EMUs → + HSR supply capacity → + Passenger turnover"

(4) "Passenger turnover → − Unit operating cost → + Target tariff → + Total transportation revenue → + Transportation profit → + the number of EMUs → + Frequency of trains → + HSRPT demand → + Passenger turnover"

(5) "Passenger turnover → + Annual operating costs → + Unit operating cost → + Target tariff → + Total transportation revenue → + Transportation profit → + the number of EMUs → + HSR supply capacity → + Passenger turnover"

(6) "Passenger turnover → + Annual operating costs → + Unit operating cost → + Target tariff → + Total transportation revenue → + Transportation profit → + the number of EMUs → + Frequency of trains → + HSRPT demand → + Passenger turnover"

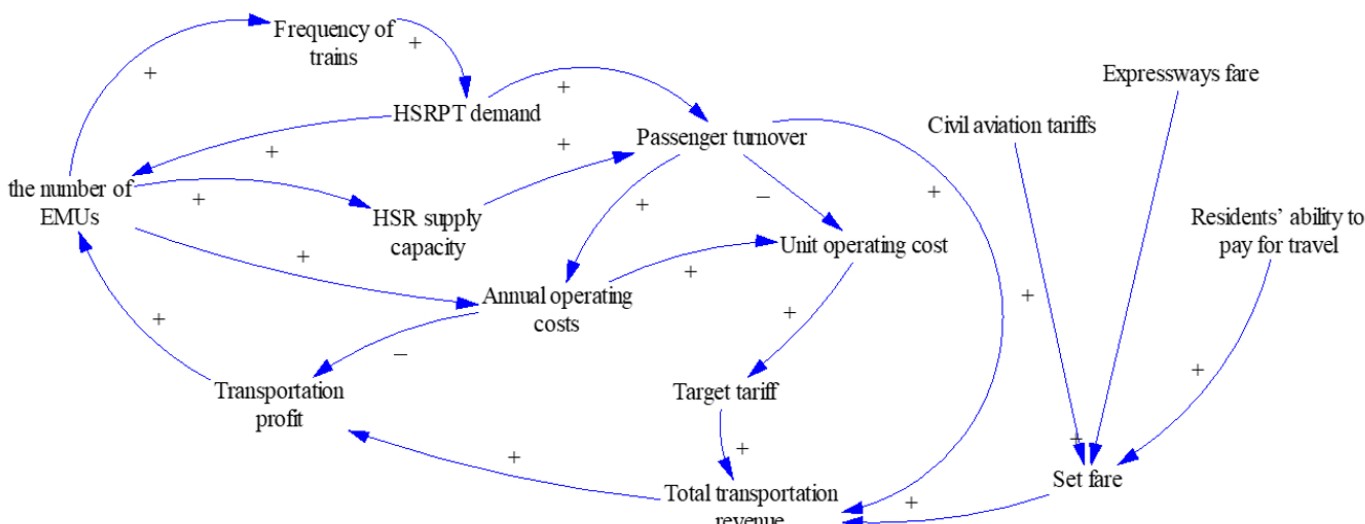

**Figure 1.** Causal loop between factors influencing the sustainable development of HSRPT under ETMM.

Loop 1 is a positive feedback loop, where an increase in passenger turnover leads to an increase in transportation revenue and profits, resulting in an increase in the number of EMUs available and in the passenger supply capacity of HSR, which is able to carry more passengers.

Loop 2 is a negative feedback loop, the increase in passenger turnover leads to an increase in the annual operating costs that need to be invested and thus a decrease in transport profits, the number of EMUs available decreases, and subsequently the frequency of trains decreases, the convenience of HSR decreases, and the demand for HSRPT decreases leading to a decrease in passenger turnover.

Loop 3 and loop 4 are negative feedback loops. Passenger turnover increases, the unit operating cost decreases accordingly, then the target tariff and total transportation revenue, transport profit will be reduced, the number of EMUs available will be reduced, the frequency of trains will also be reduced, thus decreasing the convenience of HSR service will lead to a reduction in the demand for HSRPT, and ultimately the passenger turnover is reduced.

Loop 5 and loop 6 are positive feedback loops. An increase in passenger turnover requires more operating costs to be invested, then the target tariff will increase, total transportation revenue and profit will increase accordingly, the number of EMUs available will increase, and thus the demand and supply capacity of HSRPT will increase, and thus passenger turnover will increase.

### 3.2. Model Design

According to the constructed CLD, we further analyze the quantitative relationship between various factors and build a dynamic model of the sustainable operation system of HSRPT under the ETMM. When building the model, the ETMM has the following differences compared to the general operating model.

① The annual operating cost of the ETMM includes ETM fees, EMU usage fees, road network usage fees, power supply maintenance cost, energy cost, fixed cost, R&D cost, etc. Its value is not just proportional to the passenger turnover rate, but is closely related to the number of EMUs, the number of trains, the EMU mileage, and the passenger turnover.

② The number of EMUs used in ETMM is determined not only by the purchase price of the EMU, but also by the usage fees of the EMU and the purchase price of the EMU.

Since many variables are involved in the actual operation of HSRPT, we propose the following simplifying assumptions before the model:

①: Under normal circumstances, the demand for HSPPT is increasing year by year, and we set the annual increase a fixed value.

②: We set that the number of EMU trains in operation is directly proportional to the number of EMUs.

③: Power supply maintenance costs, energy costs, R&D costs, etc. are included in the variable costs, and the annual variable costs increase proportionally.

(1) Transportation profit calculation

In Equations (1) and (2), the HSRPT demand (*D*) is gradually increasing year by year above the initial value (*ID*), the annual growth rate is set as a constant K, and the transportation profit (*TP*) also changes over time above the initial value (*ITP*). This is a cumulative amount. In Equations (3) and (4), the annual transportation profit is expressed as Profit Increasing (*PI*) for Transportation, the difference between total annual transportation revenue (*TR*) and annual operating costs (*C*), and the cost of increased demand for EMUs due to annual passenger turnover, causes a profit decrease (*PD*). Additionally, 365 * increased number of EMUs (*IN*) * EMU utilization rate ($\gamma$) represents increased numbers of EMU trains (INT), and M represents Unit vehicle operating mileage and *UVF* represents Unit EMU vehicle kilometer usage fees and $M \times UVF$ represents Unit EMU usage fees (*UUF*).

$$D = ID + tK \tag{1}$$

$$TP = ITP + t(PI - PD) \tag{2}$$

$$PI = TR - C \tag{3}$$

$$PD = 365\gamma IN \times UUF \tag{4}$$

$$UUF = M \times UVF \tag{5}$$

(2) EMU numbers calculation

In Equation (6), the increased number of EMUs (*IN*) is determined by both supply and demand. In Equations (7) and (8), the magnitude of the profit determines the number of EMUs that can be increased (*CIN*), the number of EMUs required (*RN*), the number of EMUs (*N*) and the initial value of the number of EMUs (*IE*) determines the number of EMUs that need to be increased (*NIN*). In Equation (9), *S* is the level of demand needing to be added for EMUs, and it is a constant value.

$$IN = \begin{cases} CIN, \ CIN < NIN \\ NIN, CIN \geq NIN \end{cases} \tag{6}$$

$$CIN = \frac{PI}{365\gamma UF} \tag{7}$$

$$NIN = RN - (N - IE) \tag{8}$$

$$RN = \frac{(D - ID)}{S} \tag{9}$$

(3) Operating cost calculation

In Equation (10), the annual operating cost (*C*) mainly consists of ETM fees (*MF*), EMU usage fees (*UF*), other variable cost (*VF*), fixed cost (*FF*) and Railway network usage fees (*RUF*), and the ETM fees (*MF*) consists of EMU service fees (*MSF*), equipment maintenance and station passenger service fees (*E&SF*). EMU usage fees (*UF*) are determined by vehicle operating mileage, while variable cost (*VF*) is determined by passenger turnover (*T*). The above variables are calculated as Equations (10)–(16). Where *UMSF* represents unit vehicle kilometers EMU train service fees, *UE&SF* represents unit operating mileage

equipment maintenance station passenger service fees, *NT* represents number of EMU trains, *IVNT* represents initial value of number of EMU trains, *IRUF* represents initial value of railway network usage, $\delta$ represents variable cost rate, *UC* represents unit operating cost. Additionally, *NT* equals 365 * Frequency of trains (*FNT*) in Equation (12).

$$C = MF + UF + VF + FF + R \tag{10}$$

$$MSF = NT \times M \times UMSF \tag{11}$$

$$NT = FNT \times 365 \tag{12}$$

$$MF = M \times UE\&SF + MSF \tag{13}$$

$$UF = NT \times M \times UVF \tag{14}$$

$$RUF = \frac{NT}{IVNT} \times IRUF \tag{15}$$

$$VC = \delta T \tag{16}$$

$$UC = \frac{C}{T} \tag{17}$$

(4) Passenger turnover calculation

The passenger turnover (*T*) supported by the HSR is determined by HSR supply capacity (*SC*) and HSR demand (*D*), the supply capacity of the HSR is determined by the number of EMU trains, and the demand is the HSRPT demand. Every EMU train has its supply capacity called *ASC*, the above variables are calculated as Equations (18) and (19).

$$T = \begin{cases} SC, SC < D \\ D, SC \geq D \end{cases} \tag{18}$$

$$SC = ASC \times NT \tag{19}$$

(5) Transportation revenue calculation

The total transportation revenue (*TR*) is determined by the base fare (*BF*) and the revenue from the use of the road network (*RURN*). The base fare (*BF*) is made up of the target fare (*TF*) determined by the profit rate ($\varepsilon$) and business tax rate ($\epsilon$) and the set fare (*SF*) determined by the regional residents' ability to pay for travel (*RAP*) [49,50] The importance of both is determined by parameter 1 called $\alpha$. The *SF* is determined by the residents' ability to pay for the trip (*RAP*) and the competitive fare in the market. Parameter 2 called $\beta$ determines the importance of the two, and parameter 3 called $\omega$ determines the importance of civil aviation fare (*CAF*) [50] and expressways fare (*EF*) [50]. Additionally, HSR to civil aviation fare comparison is *CCAF*, HSR to expressway fare comparison is *CEF*. *RURN* are determined by the rate of growth of *RURN* ($\mu$). Interval fare for civil aviation and expressways are linked to time and the annual interest rate of current demand deposits ($\rho$). The above variables are calculated as Equations (20)–(27).

Where *IRURN* represents initial value of RURN, $\pi$ is growth rate of *RURN*, *NRAP* is national residents' ability to pay for travel, $\theta$ is comprehensive coefficient of regional living standards.

$$TR = T \times BF + RURN \tag{20}$$

$$RURN = IRURN \times (1 + \pi)^{(t-2019)} \tag{21}$$

$$BF = \alpha \times TF + (1 - \alpha) \times SF \tag{22}$$

$$TF = UC \times \frac{1 + \varepsilon}{1 - \epsilon} \tag{23}$$

$$SF = \beta \times RAP + (1 - \beta) \times [\omega \times CAF \times CCAF + (1 - \omega) \times EF \times CEF] \tag{24}$$

$$RAP = NRAP \times \theta \tag{25}$$

$$CAF = ICAF \times e^{(t-2019) \times LN(1+\rho)} \tag{26}$$

$$EF = IEF \times e^{(t-2019) \times LN(1+\rho)} \tag{27}$$

Based on the construction of the above expression, the corresponding stock flow diagram is drawn using VENSIM software, as shown in Figure 2. The arrows in the figure point to both ends to indicate variables linked with expression relations.

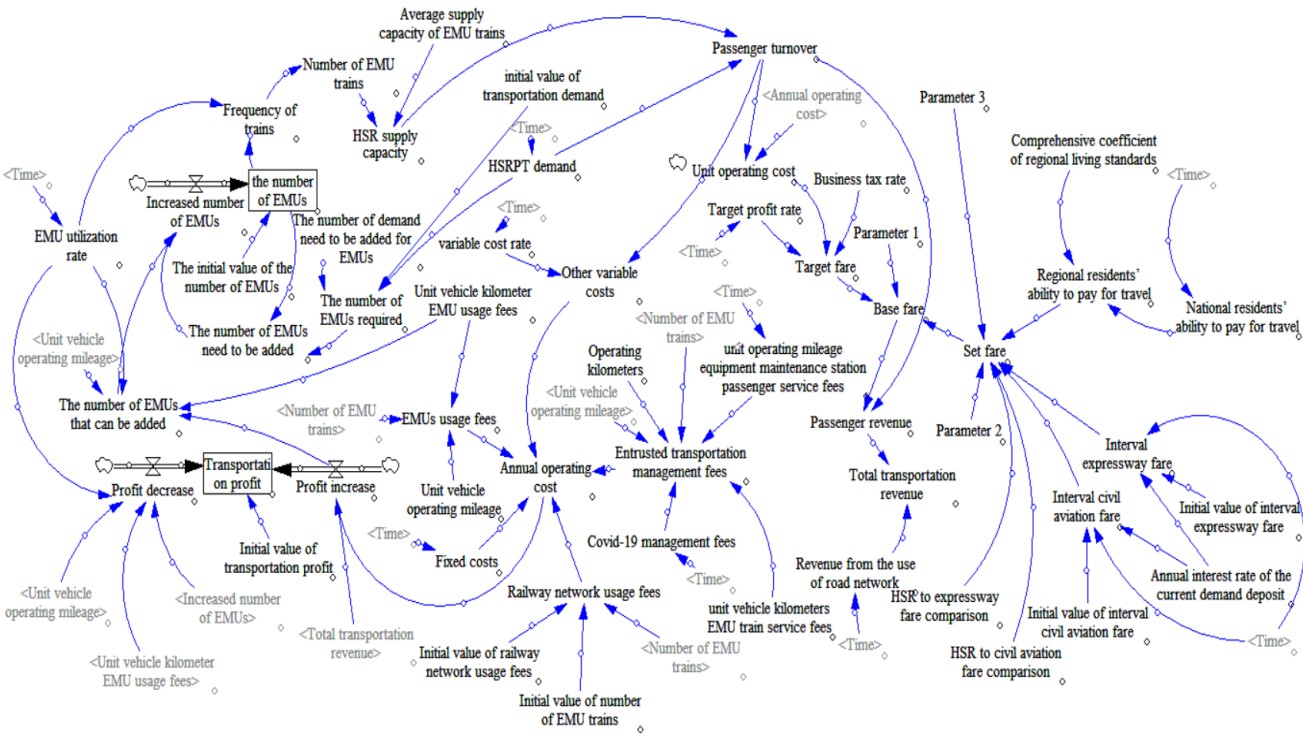

**Figure 2.** Dynamics flow diagram of sustainable operation system of HSRPT under ETMM. ◇ is equivalent to mirror variable.

## 4. Case Study

The Beijing–Shanghai HSR runs through the three municipalities directly under the Central Government of Beijing, Tianjin and Shanghai and the four provinces of Hebei, Shandong, Anhui and Jiangsu, connects the two major economic zones of "Beijing-Tianjin-Hebei" and "Yangtze River Delta", provides "safe, fast, convenient and comfortable" HSRPT services, has a strong competitiveness in the Beijing–Shanghai passenger transport market.

The Beijing–Shanghai HSR adopts ETMM. The following uses the Beijing–Shanghai HSR as an example to study the sustainable development mechanism of HSRPT under the ETMM using the above-mentioned SD model.

### 4.1. Data

In order to better study the dynamic changes of the system, the time interval of the model is set to one year. Due to the impact of COVID-19 in 2020, the operation of the Beijing–Shanghai HSR is abnormal, and passenger turnover, transportation revenue, and transportation profits have all been affected, resulting in greater fluctuations. In order to study the normalization and sustainable operation mechanism of the ETMM, the initial year is set to 2018, and the end year is set to 2023. The data comes from the "Beijing-Shanghai HSR Co., Ltd. Annual Report 2019" [51] and the "Beijing-Shanghai HSR Co., Ltd. Initial Public Offering Prospectus" [1]. The specific parameter values are obtained from the following channels:

(1) Some parameters can be obtained directly from the annual report data of Beijing–Shanghai HSR over the years, such as the initial value of Beijing–Shanghai HSR

transportation demand, the initial value of revenue from the use of road network, and Unit EMU usage fees.

(2) Some parameters need to be deduced and calculated through historical data, such as the increase in transportation demand, fixed cost value, and the growth rate of road network revenue.

(3) Some parameters require regression analysis to obtain future data based on historical data, such as the national residents' travel affordability and the comprehensive coefficient of living standards in the east.

(4) Some parameters need to be obtained through SD simulation results, such as transportation profit, transportation revenue, basic fare, etc.

The model input data is shown in Table 1 below:

**Table 1.** Input parameter values.

| Parameter | 2018 | 2019 | Unit |
|---|---|---|---|
| The initial value of Beijing–Shanghai HSR transportation demand ($D$) | 344.57 | 346.71 | Billion person kilometers |
| Increase in transportation demand ($ID$) | 9.49 | 9.49 | Billion person kilometers |
| Initial value of the number of EMUs ($IE$) | 0.144 | 0.163 | Thousand |
| Initial value of transportation profit ($IP$) | 0 | 151.14 | 100 million yuan |
| EMU utilization rate ($\gamma$) | 75 | 75 | % |
| Unit vehicle operating mileage ($M$) | 1318 | 1318 | Kilometer |
| Unit EMU usage fees ($UUF$) | 0.80 | 0.80 | Ten thousand yuan/thousand kilometers |
| The level of demand needing to be added for EMUs ($S$) | 2.1 | 2.1 | Billion person kilometers |
| Unit vehicle kilometers EMU train service fees ($UMSF$) | 0.12 | 0.12 | Ten thousand yuan/thousand kilometers |
| Unit operating mileage equipment maintenance and station passenger service fees | 181.26 | 181.26 | Ten thousand yuan/operating kilometer |
| Variable cost rate ($UE\&SF$) | 0.14 | 0.14 | Yuan/person km |
| Fixed cost ($FF$) | 55.06 | 55.06 | 100 million yuan |
| Average supply capacity of EMU trains ($ASC$) | 0.0157 | 0.0157 | 100 million person kilometers/column |
| Initial value of RURN ($IRURN$) | 151.02 | 170.18 | 100 million yuan |
| Growth rate of RURN ($\pi$) | 12.65 | 12.65 | % |
| Target profit rate ($\varepsilon$) | 5 | 5 | % |
| Business tax rate ($\epsilon$) | 5 | 5 | % |
| National residents' ability to pay for travel ($NRAP$) | 0.62 | 0.65 | Yuan/km |
| Comprehensive coefficient of regional living standards ($\theta$) | 129 | 129 | % |
| Initial value of interval civil aviation fare ($ICAF$) | 1.36 | 1.36 | Yuan/km |
| HSR to civil aviation fare comparison ($CCAF$) | 30 | 30 | % |
| Initial value of interval expressway fare ($IEF$) | 0.81 | 0.81 | Yuan/km |
| HSR to expressway fare comparison ($CEF$) | 51 | 51 | % |
| Annual interest rate of current demand deposit ($\rho$) | 0.35 | 0.35 | % |
| $\alpha$ | 7 | 7 | % |
| $\beta$ | 5 | 5 | % |
| $\omega$ | 5 | 5 | % |

Among them, NRAP = (0.62, 0.65, 0.65, 0.68, 0.71, 0.75).

### 4.2. Simulation Results

#### 4.2.1. Model Validation

Using real data to simulate the sustainable development of Beijing–Shanghai HSR passenger transportation under the ETMM. In the sustainable development indicators, the operating indicators use passenger turnover, and the economic indicators use total transportation revenue and transportation profits. Based on the availability and completeness of the data, the data of the Beijing–Shanghai HSR in 2018 and 2019 are used to verify the validity of the SD model. The simulation results are shown in Table 2 below.

As can be seen from the table, the errors of total transportation revenue, operating costs, transportation profits, and passenger turnover in 2018 and 2019 are basically within 5%, of which the error of transportation profit in 2018 is 1.72%, and the error of transportation

profit in 2019 is 2.79 %, passenger turnover is 0%, the operating cost error in 2018 is 4.39%, and the operating cost error in 2019 is 6.35%. Generally speaking, the simulation results of the SD model are within the allowable error range, and the model is effective reasonable.

**Table 2.** Model validation.

| Time | | *TR*/100 million yuan | *C*/100 million yuan | *TP*/100 million yuan | *T*/100 million person kilometers |
|------|------|------|------|------|------|
| | Actual Value | 311.58 | 162.99 | 148.59 | 344.57 |
| 2018 | Analog Value | 321.28 | 170.14 | 151.14 | 344.57 |
| | Error | 3.11% | 4.39% | 1.72% | 0% |
| | Actual Value | 326.64 | 161.03 | 165.61 | 346.71 |
| 2019 | Analog Value | 341.49 | 171.25 | 170.24 | 346.71 |
| | Error | 4.55% | 6.35% | 2.79% | 0% |

The reasons for the errors are:

① In the operating costs, R&D costs, power supply maintenance fees, energy fees, and other costs are attributed to variable costs. In the model, variable costs are set to a value that is proportional to the passenger turnover. The actual operating cost calculation process is more complicated. The simulation effect will be different.

② The value of the target profit rate determines the basic fare and thus the passenger transportation revenue. The target profit rate can only be estimated within a reasonable range.

③ The number of EMU trains are determined by the number of EMU and the utilization rate of EMU. This method can determine the number of trains. However, the actual operation process is complicated. The number of EMU trains is related to the preparation of train operation plans, so there are certain differences.

4.2.2. Simulation of the Sustainable Development Status of Beijing–Shanghai HSR

The SD model is used to simulate the sustainable development of Beijing–Shanghai HSR passenger transportation under the ETMM. It is used to simulate the operating parameters of the Beijing–Shanghai HSR from 2018 to 2023 under normal conditions (without the epidemic), and to analyze the sustainability of the development of the Beijing–Shanghai HSR under the ETMM.

The changing trend of passenger turnover is shown in Figure 3 below. With the growth of the years, residents' demand for passenger transportation has also increased year by year. Since the supply capacity of the Beijing–Shanghai HSR exceeds the growth in demand, passenger turnover has shown an upward trend year by year.

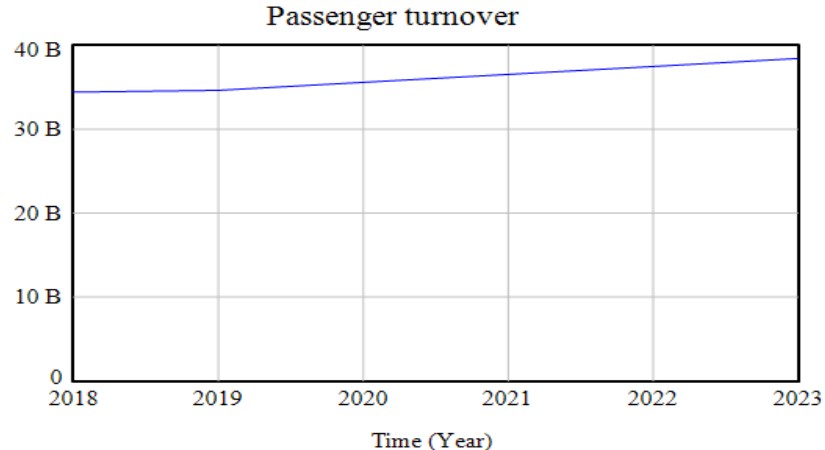

**Figure 3.** Changing trend of passenger turnover.

The simulation results of the total transportation revenue and related parameters are shown in Figure 4. Under the ETMM, the total transportation revenue is divided into the passenger transportation revenue of the line and the revenue from the use of the road network. Passenger revenue is determined by passenger turnover and basic fares (without epidemic situation), the basic fares have little fluctuations, and the growth of passenger revenue on this line is not large. This is due to the passenger transportation revenue of the Beijing–Shanghai main line, which has only shown a slow growth trend in recent years. The comprehensive passenger load factor of the Beijing–Shanghai route has been stable at around 80% all year round. With limited room for passenger load factor to increase, coupled with fierce civil aviation competition, ticket revenue will grow slowly or even close to the same state in the future by 2023. The revenue from the use of road networks has increased relatively rapidly. With the improvement of the road network usage fees system, revenue from the use of cross-line networks may become a larger proportion of revenue growth in the future.

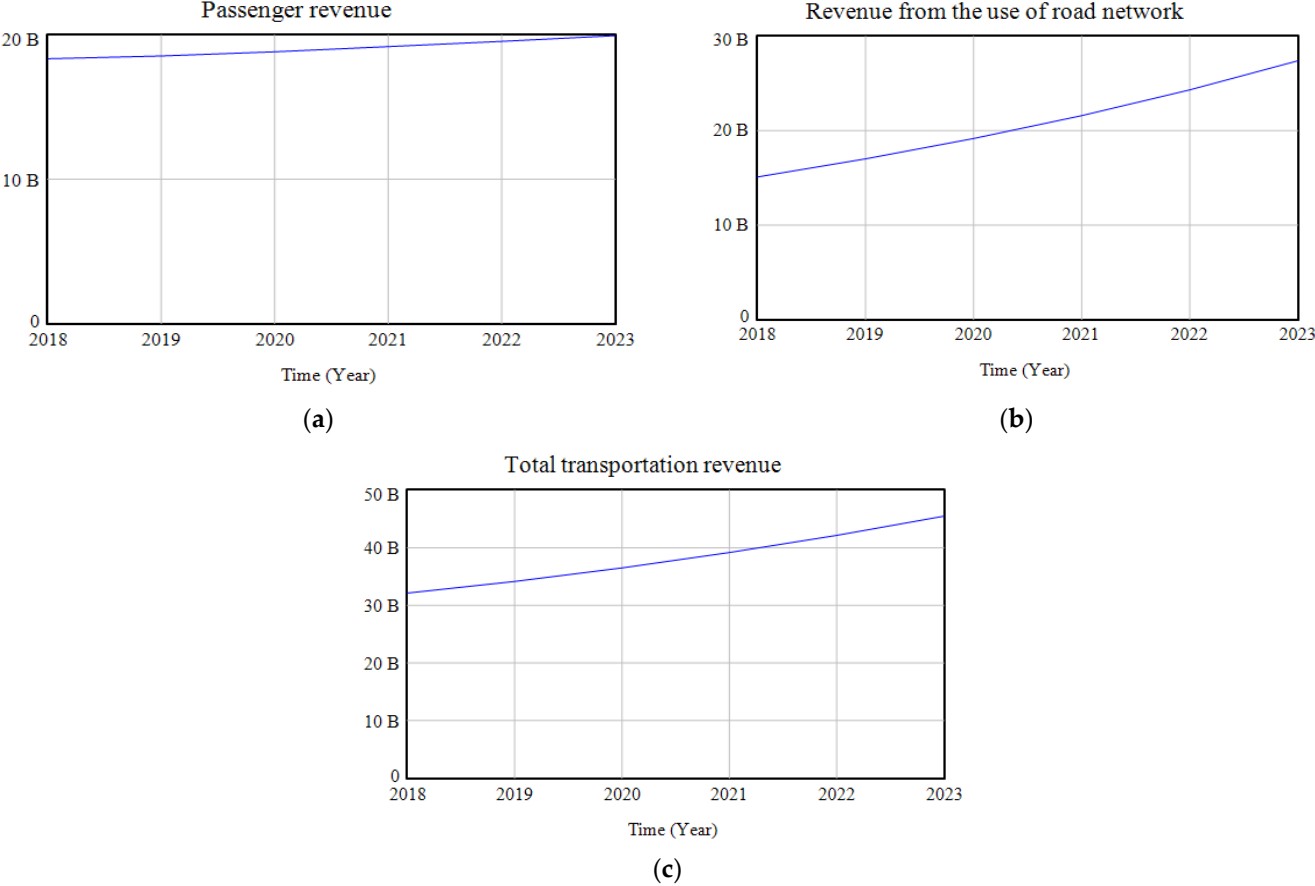

**Figure 4.** Trend of transportation revenue: (**a**) passenger revenue; (**b**) revenue from the use of road network; (**c**) total transportation revenue.

The simulation results of operating costs and related parameters are shown in Figure 5. Under the ETMM, with the increase in passenger turnover, the corresponding ETM fees increase and the increase in transportation demand results in the increasing number of trains of EMU. With the increase, EMU usage fees and road network usage fees have also increased correspondingly, and annual operating costs have shown an increasing trend.

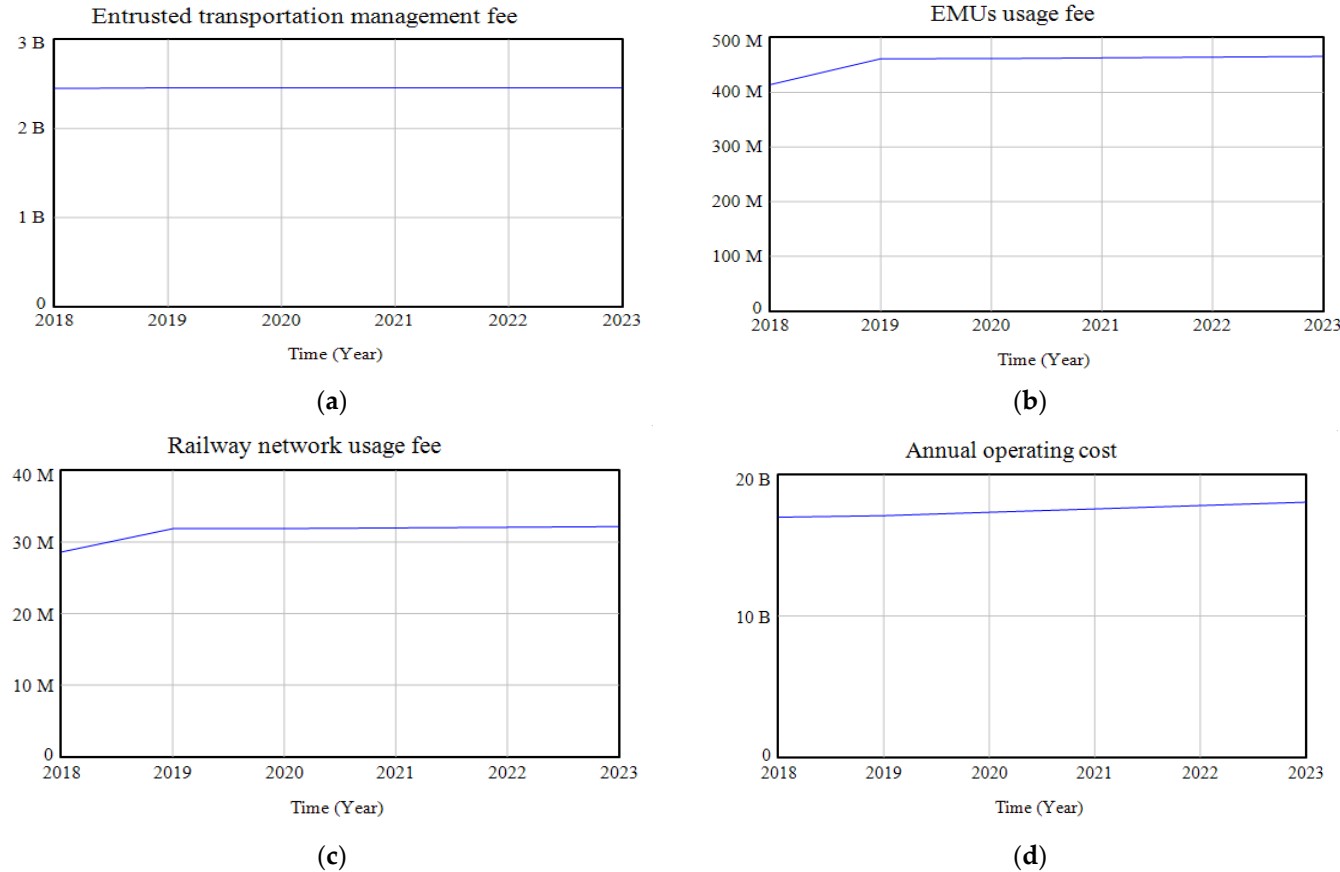

**Figure 5.** Trend of operating cost: (**a**) entrusted transportation management fees; (**b**) EMUs usage fees; (**c**) railway network usage fees; (**d**) annual operating cost.

The simulation results of transportation profit and related parameters are shown in Figure 6. The transportation profit at the beginning of each year is the sum of the transportation profits of the previous years. The profit growth represents the annual transportation profit. With the growth of the year, the annual transportation profit increases.

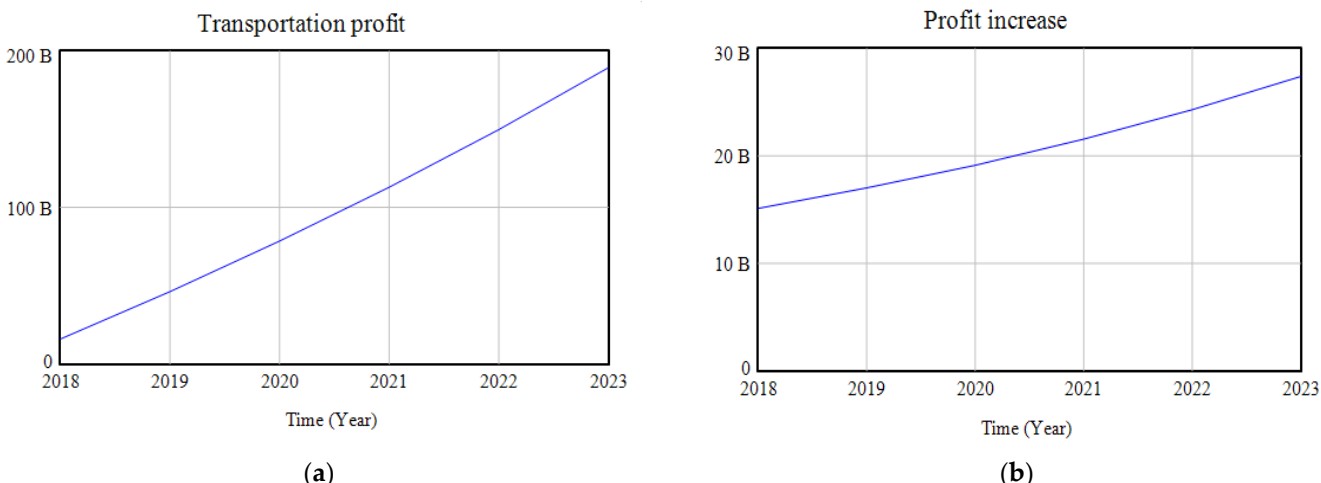

**Figure 6.** Trend of transportation profit: (**a**) transportation profit; (**b**) profit increase.

In summary, it can be seen that, in terms of economic benefits, the Beijing–Shanghai HSR runs sustainably.

The results obtained are summarized in Table 3 below, in which the influence trend of passenger turnover on transportation revenue and transportation profit is consistent with the constructed SD causality, indicating that the sustainable operation model based on system dynamics is effective.

**Table 3.** Summary of simulation results.

| Time | TR | Passenger TR | RURN | C | MF | UC | T | TP |
|---|---|---|---|---|---|---|---|---|
| Unit | 100 million yuan | 100 million yuan | 100 million yuan | 100 million yuan | 100 million yuan | yuan | 100 million person kilometers | 100 million yuan |
| 2018 | 321.28 | 183.65 | 151.02 | 170.14 | 245.11 | 0.494 | 344.57 | 151.14 |
| 2019 | 341.49 | 185.54 | 170.18 | 171.25 | 245.82 | 0.494 | 346.71 | 170.24 |
| 2020 | 364.94 | 188.43 | 191.71 | 173.63 | 245.83 | 0.487 | 356.20 | 191.31 |
| 2021 | 391.69 | 192.05 | 215.96 | 176.02 | 245.85 | 0.481 | 365.69 | 215.67 |
| 2022 | 421.42 | 195.70 | 243.28 | 178.41 | 245.87 | 0.475 | 375.18 | 243.01 |
| 2023 | 454.74 | 199.64 | 274.05 | 180.80 | 245.89 | 0.470 | 384.67 | 273.94 |

*4.3. Sensitivity Analysis*

From the above analysis, it can be seen that the annual demand for HSR transportation plays a crucial role in the sustainable operation system of HSR. With the increase in the demand for transportation, transportation profit, transportation income and transportation cost are changing. Therefore, it is necessary to study the impact of transportation demand variation on the system. In addition, compared with the general operation mode, the most significant change of ETMM is the change of operation cost structure, so it is necessary to study the impact of entrusted transportation management cost on the system.

4.3.1. Sensitivity Analysis of Transport Demand

By changing the increment of passenger transport demand and studying the changes of inter-system variables and their impact on the sustainable operation of HSR, we can adapt to market changes and adjust the operation strategy. In this paper, the growth of transportation demand is set as 500 million person km, 1 billion person km, 1.5 billion person km and 2 billion person km, respectively, for simulation. The changes of unit operating cost, total operating cost and transportation profit are shown in Figure 7.

It can be seen from Figure 7 that, under the ETMM, the service of the Beijing–Shanghai HSRPT is undertaken by commissioned railway transport enterprises along the line, the number of EMU is determined by the annual transport profit and the EMU fees per unit vehicle mileage. With the growth of transportation demand, EMU fees per unit vehicle mileage is fixed, annual operating cost increases, and unit operating cost decreases correspondingly, so the basic ticket price decreases slightly, and the increase in transportation profits is modest.

4.3.2. Sensitivity Analysis of Entrusted Transportation Management Fees

By changing entrusted transportation management fees, it has a direct impact on operating costs. Changes in operating costs will also affect transport profits and passenger transport capacity. In this paper, unit-operating mileage equipment maintenance and station passenger service fees are set as fixed values and time-varying variables for simulation. The trend of total transportation revenue and unit operating cost is shown in Figure 8.

With the change in the entrusted transportation management fees, annual operating costs increased with the increase in unit-operating mileage equipment maintenance and station passenger service fees. While passenger turnover was unchanged, unit operating costs increased. However, the total operating revenue increased with the increase in entrusted transportation management fees, and the operating income still increased because of the adjustment of basic rates. In this way, by adjusting ticket prices within a reasonable

range, the economic benefits of the Beijing–Shanghai HSR will be basically stable and sustainable development will be achieved.

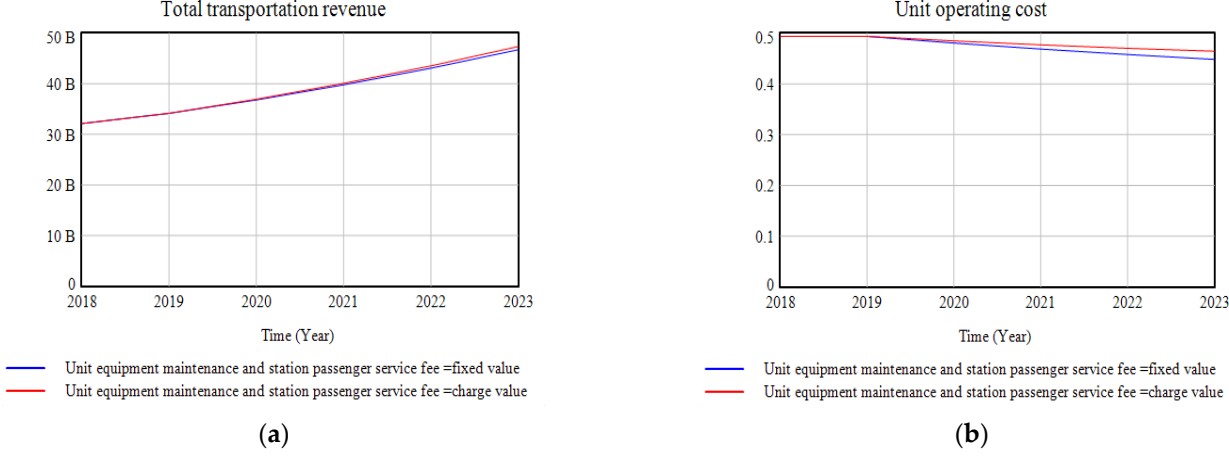

**Figure 7.** Changes of (**a**) unit operating cost; (**b**) total operating cost; (**c**) transportation profit.

**Figure 8.** The trend of (**a**) total transportation revenue; (**b**) unit operating cost.

*4.4. COVID-19 Impact Simulation*

In 2020, COVID-19 had a huge impact on the development of Beijing–Shanghai HSRPT, so it is necessary to analyze the sustainable development status of the Beijing–Shanghai HSR. From the qualitative perspective, the impact of COVID-19 on Beijing–Shanghai HSRPT is mainly reflected in the following aspects:

(1) Passenger turnover is greatly reduced. The Beijing–Shanghai HSR carried 18.098 billion passenger kilometers in 2020, 47.8 percent lower than in 2019.

(2) Entrusted transportation management fees increased. As the Beijing–Shanghai HSR Company takes epidemic prevention and control measures, the cost invested by the entrusted Transport railway Bureau for station maintenance and management increases, so do the entrusted transport management fees paid by the Beijing–Shanghai HSR.

(3) Revenue from the use of network slows down accordingly. Railway passenger transport across the country has been affected by the epidemic, and the revenue from the use of network of trains passing through the Beijing–Shanghai HSR carried by other railway transport companies have also been reduced accordingly. Revenue from the use of network in 2020 is 17.23 billion yuan, while the revenue from the use of network in 2020 is expected to be 19.171 billion yuan without the impact of the epidemic, a corresponding decrease of 10.13%.

(4) Train frequency reduction. As passenger transport demand decreases, the Beijing–Shanghai high-speed Railway will reduce the number of EMU trains issued to reduce costs. As the ownership of EMU remains unchanged, the utilization rate of EMU will also decrease, and the frequency of trains will decrease accordingly.

From the perspective of SD, in order to study the impact of epidemic situations of different severity on the sustainable operation of the Beijing–Shanghai HSR, this paper proposes the concept of epidemic impact factor $\alpha$, and assumes that the epidemic impact factor $\alpha$ in 2020 is 1. As the regions along the Beijing–Shanghai HSR are economically developed and densely populated, and the main part of the passenger flow is business passenger flow, the impact of the epidemic on the passenger flow of the Beijing–Shanghai HSR is gradually reduced as COVID-19 is gradually brought under control. Assume that the epidemic impact factor $\alpha = 0.7$ in 2021, 0.2 in 2022, and 0 in 2023. Introducing epidemic prevention and control cost, it cost 11 million yuan in 2020. Adjustments to other parameters are shown in Table 4 below.

**Table 4.** Adjust parameter input.

| | Unit | 2020 | 2021 | 2022 | 2023 |
|---|---|---|---|---|---|
| COVID-19 management cost | ten thousand yuan | 1100 | 770 | 220 | 0 |
| Road network use revenue | hundred million yuan | 172.29 | 200.60 | 238.35 | 268.50 |
| Unit entrusted transportation management fees | ten thousand yuan | 203.81 | 213.62 | 221.40 | 235.79 |
| Utilization rate of EMU ($\gamma$) | % | 40 | 50.5 | 74 | 75 |

Entering the above data and simulating with VENSIM software, we have obtained the main data summary table of the Beijing–Shanghai HSR operation under different epidemic impacts, as shown in Table 5 below.

Main data of the Beijing–Shanghai HSR operation under the influence of COVID-19 are shown in Figure 9 below.

It can be seen from Table 5 and Figure 9 that, in 2020, due to the impact of COVID-19, passenger transport demand decreased, and passenger turnover declined sharply in 2020. After 2020, with the decrease in the severity of COVID-19, passenger turnover showed an upward trend, and it is basically the same as that in 2019 around 2022.

Compared with 2019, transportation profit decreased significantly. After 2020, the impact of COVID-19 decreased, transportation profit gradually increased. In 2022, it is basically the same as that in 2019, and returned to normal in 2023.

**Table 5.** Summary table of operation data of the Beijing–Shanghai HSR under the impact of COVID-19.

| Time | Circumstances | TR | Passenger TR | RURN | C | MF | UC | TP | T |
|---|---|---|---|---|---|---|---|---|---|
| **Unit** | *I* | hundred million yuan | hundred million yuan | hundred million yuan | hundred million yuan | hundred million yuan | yuan | hundred million yuan | hundred million passenger-kilometer |
| 2020 | I | 364.94 | 188.43 | 191.71 | 173.63 | 245.83 | 0.487 | 191.31 | 356.20 |
| | II | 247.84 | 85.87 | 172.29 | 198.70 | 247.01 | 1.098 | 49.14 | 180.98 |
| | I − II | 117.10 | 102.56 | 19.42 | −25.07 | −1.18 | −0.61 | 142.17 | 175.22 |
| 2021 | I | 391.69 | 192.05 | 215.96 | 176.02 | 245.85 | 0.481 | 215.67 | 365.69 |
| | II | 317.54 | 130.12 | 200.65 | 198.35 | 248.30 | 0.827 | 119.18 | 239.77 |
| | I − II | 74.15 | 61.93 | 15.31 | −22.33 | −2.45 | −0.35 | 96.49 | 125.92 |
| 2022 | I | 421.42 | 195.70 | 243.28 | 178.41 | 245.87 | 0.475 | 243.01 | 375.18 |
| | II | 390.91 | 168.85 | 238.35 | 195.51 | 251.87 | 0.578 | 195.41 | 338.27 |
| | I − II | 30.51 | 26.85 | 4.93 | −17.10 | −6.00 | −0.10 | 47.60 | 36.91 |
| 2023 | I | 454.74 | 199.64 | 274.05 | 180.80 | 245.89 | 0.470 | 273.94 | 384.67 |
| | II | 459.32 | 204.41 | 268.50 | 185.78 | 252.03 | 0.483 | 273.54 | 384.67 |
| | I − II | −4.58 | −4.77 | 5.55 | −4.98 | −6.14 | −0.01 | 0.40 | 0.00 |

I: Index value without epidemic impact. II: Index values under the impact of the epidemic.

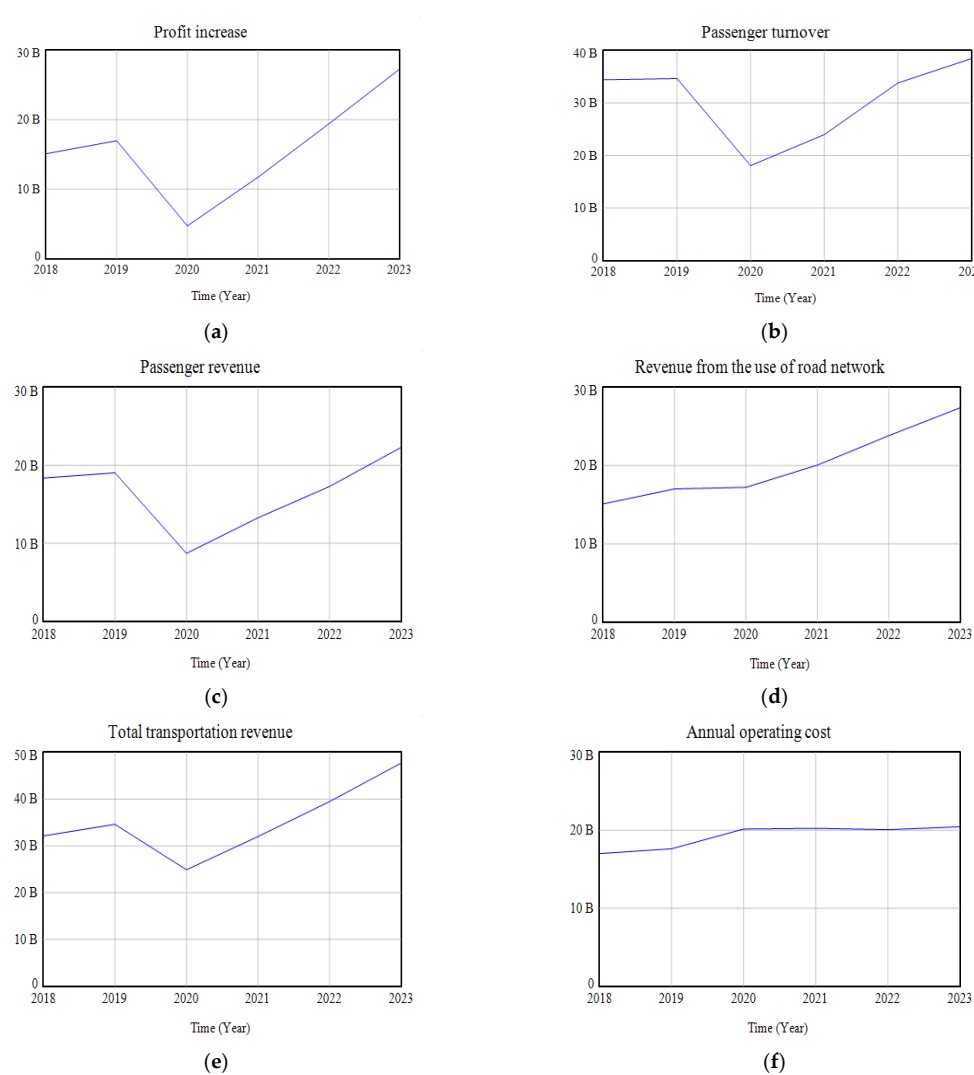

**Figure 9.** Simulation results under the impact of the pandemic: (**a**) profit increase; (**b**) passenger turnover; (**c**) passenger revenue; (**d**) revenue from the use of network; (**e**) total transportation revenue; (**f**) annual operating cost.

Passenger revenue is relatively low, and similarly, revenue from the use of network have been declining, but the impact is less than that of passenger revenues. After 2020, with the decrease in the severity of COVID-19, the revenue of passenger transportation and the revenue from the use of network is gradually increasing. In 2023, the revenue of passenger transportation will be equal to that of 2019.

Although the growth rate of road network use revenue slowed down due to the impact of COVID-19, it still showed an increasing trend compared with 2019. Operating costs increased in 2020 compared with 2019. After 2020, the severity of COVID-19 decreased, and the growth rate of annual operating costs slowed down.

## 5. Discussion

Without considering COVID-19, the annual growth rate of the simulation results of SD in Table 6 shows that the sustainable development capacity of the Beijing–Shanghai HSR keeps a steady improvement year by year. The indicators of total transportation revenue, annual operating costs, transportation profit and passenger turnover et al. Table 6 show a strengthening trend of development. In the past six years, the rate of transportation revenue and transportation profit growth rate has been constantly improving. Transportation revenue will reach 8% in 2023, and transportation profit growth rate will reach 12.73%. The passenger turnover increases year by year, but the unit operating costs decreases. Unit operating costs will drop to 0.470 in 2023, which means the Beijing–Shanghai HSR will generate more revenue at lower costs. The increase in profit and the decrease in operating costs indicate that the passenger transport of the Beijing–Shanghai HSR under the entrusted management operation mode has good sustainability.

**Table 6.** The simulation results increase year by year.

| Time | Growth Rate of *TR* | Growth Rate of Passenger *TR* | Growth Rate of *RURN* | Annual Growth Rate of *C* | Growth Rate of *MF* | Growth Rate of *UC* | Growth Rate of *TP* | Growth Rate of *T* |
|------|------|------|------|------|------|------|------|------|
| 2019 | 6.29 | 1.03 | 12.69 | 0.65 | 0.29 | 0 | 12.57 | |
| 2020 | 6.87 | 1.56 | 12.65 | 1.39 | 0.01 | −0.01 | 12.38 | 2.73 |
| 2021 | 7.33 | 1.92 | 12.64 | 1.38 | 0.01 | −0.01 | 12.73 | 2.66 |
| 2022 | 7.59 | 1.90 | 12.65 | 1.36 | 0.01 | −0.01 | 12.68 | 2.59 |
| 2023 | 7.91 | 2.01 | 12.64 | 1.34 | 0.01 | −0.01 | 12.73 | 2.52 |

In 2020, due to the impact of COVID-19, the Beijing–Shanghai HSR broke the normal operation, and passenger service was impacted. The change ratio of industry indicators under the influence of COVID-19 and those without the influence of COVID-19 is shown in Table 7. According to the simulation results of SD, the industry indexes of the Beijing–Shanghai HSR under the influence of COVID-19 and the normal situation gradually decreases in 2020–2022. In 2023, COVID-19 basically has no impact on the operation of the Beijing–Shanghai HSR, and there is little difference between various industry indicators and the normal situation.

**Table 7.** Comparison of changes in indicators under the influence of the epidemic from 2020–2023.

| Time | *TR* Ratio | Passenger *TR* Ratio | *RURN* Ratio | *C* Ratio | *MF* Ratio | *UC* Ratio | *TP* Ratio | *T* Ratio |
|------|------|------|------|------|------|------|------|------|
| 2020 | 0.68 | 0.46 | 0.90 | 1.14 | 1.00 | 2.25 | 0.26 | 0.51 |
| 2021 | 0.81 | 0.68 | 0.93 | 1.13 | 1.01 | 1.72 | 0.55 | 0.66 |
| 2022 | 0.93 | 0.86 | 0.98 | 1.10 | 1.02 | 1.22 | 0.80 | 0.90 |
| 2023 | 1.01 | 1.02 | 0.98 | 1.03 | 1.02 | 1.03 | 1.00 | 1.00 |

As can be seen from Table 7, in terms of total transportation revenue, the total transportation revenue from 2020 to 2023 affected by COVID-19 accounted for 67.91%, 81.07%, 92.76% and 101.00%, respectively, of the total transportation revenue unaffected by COVID-19. From the perspective of passenger transport revenue, the passenger transport revenue from

2020 to 2023 affected by COVID-19 accounted for 45.57%, 67.75%, 86.28% and 102.39%, respectively, of those unaffected by COVID-19. From the perspective of transportation profit, transportation profit from 2020 to 2023 affected by COVID-19 accounted for 25.69%, 55.26%, 80.41% and 99.85%, respectively, of those unaffected by COVID-19. From the perspective of passenger turnover, the passenger turnover from 2020 to 2023 affected by COVID-19 accounted for 50.81%, 65.57%, 90.16% and 100%, respectively, of those unaffected by COVID-19. The proportion of total transportation revenue, passenger transportation revenue, transportation profit and passenger turnover gradually increased mainly because the impact of COVID-19 gradually decreased, people returned to normal life, and the demand for travel increased.

From the perspective of road network use income, the road network use income from 2020 to 2023 affected by COVID-19accounted for 89.87%, 92.88%, 97.97% and 97.97%, respectively, of those unaffected by COVID-19. The main reason for the low impact of COVID-19 is that the main service object of the Beijing–Shanghai Railway is the Beijing–Shanghai HSR, and the use of cross-line trains is relatively small.

From the perspective of annual operating costs, the annual operating costs from 2020 to 2023 affected by COVID-19 accounted for 114.44%, 112.69%, 109.58% and 102.75%, respectively, of those unaffected by COVID-19. From the perspective of ETM fees, the ETM fees from 2020 to 2023 affected by COVID-19 accounted for 100.48%, 101.00%, 102.44% and 102.50%, respectively, of those unaffected by COVID-19. From the perspective of unit operating costs, the unit transportation costs from 2020 to 2023 affected by COVID-19 accounted for 225.46%, 171.93%, 121.68% and 102.77%, respectively, of those unaffected by COVID-19. The annual operating costs, ETM fees and unit transportation costs increased, while number of EMUs in operation decreased, mainly due to the increase in the cost of human and material resources, caused by COVID-19.

On the whole, industry production indicators such as transport revenue, transport profit, operating costs and passenger turnover under the impact of COVID-19 remain within an acceptable range. Additionally, the sustainable development capacity shows a stable trend as the impact of the epidemic gradually decreases.

## 6. Conclusions

In order to promote the sustainable and healthy development of China's railway transport industry, the government attaches great importance to the market-oriented reform of railway industry, and the application of ETMM is an important measure. This paper takes the Beijing–Shanghai HSR as an example to build an SD model of ETMM for HSR passenger transport, which can be operated sustainably. This paper evaluates the SO of the Beijing–Shanghai HSR by two indicators of economic benefit and operational benefit, and it discusses the impact of COVID-19 on the operation of the Beijing–Shanghai HSR. The research results are as follows: (1) The SD model constructed in this paper can effectively simulate the SO mechanism of HSRPT under ETMM. Additionally, it can help HSR operating companies analyze and respond to market changes and adjust their operation strategies. (2) Without the influence of COVID-19, the passenger turnover rate of Beijing–Shanghai HSR increases year by year with the increase in passenger transport demand. The total revenue and profit of transportation also increase year by year, and the growth rate also expands gradually. In 2023, the growth rate of transportation profit will reach 12.73%. (3) By analyzing the growth of passenger demand and the influence of ETM fees on the stability of the system, it can be seen that when the growth of passenger demand increases, the annual operating costs increases, but the unit operating costs decrease. By reasonably adjusting the freight rate, the operation of the Beijing–Shanghai HSR will not have large fluctuations. In the case of the sudden increase in ETM fees, the way of adjusting freight rates to a small extent to ensure that transportation profits will not fall. (4) Under the influence of COVID-19, Beijing–Shanghai HSR has been greatly impacted. Passenger turnover and transportation profit decreased by 74.31% and 49.19%, respectively, in 2020, without the epidemic situation. With the prevention and control of COVID-19, ETM fees

gradually decreased, and passenger turnover gradually returned to normal. Transport profits will gradually recover to 2019 levels in 2022. In general, the sustainable development capacity of the Beijing–Shanghai HSR under ETMM is relatively stable in the short and long term. So, the Beijing–Shanghai HSR can cope with various emergencies and realize economic and operational benefits at the same time.

There are still some defects in this paper that can be further improved. When building the SD model in future research, regional GDP factors can be considered to be integrated into the relationship to study the impact of regional economy on the SO of HSR. In addition, the paper can also consider using quantitative expressions to simulate the relationship between COVID-19 and passenger turnover, so as to make the research more objective.

**Author Contributions:** Conceptualization, W.L.; methodology, C.J. and Y.Y.; analysis, C.J., Y.Y. and J.D.; writing—original draft preparation, C.J., Y.Y. and J.D.; writing—review and editing, J.D. and Y.Y.; visualization, C.J.; supervision, W.L.; funding acquisition, W.L. All authors have read and agreed to the published version of the manuscript.

**Funding:** This research was funded by the National Key Research and Development Program, grant number 2017YFE0134600.

**Institutional Review Board Statement:** Not applicable.

**Informed Consent Statement:** Not applicable.

**Data Availability Statement:** This study used publicly available data; see http://www.cr-jh.cn/websiteMenu/213/2, accessed on 19 November 2019, PDF document of IPO prospectus of Beijing–Shanghai High-Speed Railway Co., Ltd.

**Acknowledgments:** The authors would like to acknowledge all experts' contributions in the building of the model and the formulation in this study.

**Conflicts of Interest:** The authors declare no conflict of interest.

## Abbreviations

The following abbreviations are used in this manuscript

| | |
|---|---|
| HSR | High-speed railway |
| HSRPT | High-speed rail passenger transport |
| SD | System dynamics |
| ETMM | Entrusted transportation management mode |
| ETM | Entrusted transportation management |
| CLD | Causal Loop Diagrams |
| RURN | Revenue from the use of road network |
| SO | Sustainable operation |

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
