# Peer review of "Sustainable Mechanism of the Entrusted Transportation Management Mode on High-Speed Rail and the Impact of COVID-19: A Case Study of the Beijing–Shanghai High-Speed Rail"

_sustainability, doi:10.3390/su14031171_

Round 1

Reviewer 1 Report

Dear Authors, 

Thank you for your exciting manuscript, but I have some comments:

  • In my opinion, it is not necessary to split your Introduction part into subcategories.
  • What are from 1 to 30 - formulas? Maybe you can use the letters (with explanations), not whole words.
  • please explain the fig. 1-6.

Reviewer 2 Report

The manuscript analyses the sustainable development mechanism of the Entrusted Transportation Management Mode of high-speed rail by using system dynamics method, to simulate the impact of COVID-19 on sustainable operation of the Beijing-Shanghai high-speed rail. The goal was to help companies with their sustainable operation strategies through a case study to verify the feasibility of the model. The article gives a contribution to the research and the model can be applied. The methodology is clearly explained   and theoretically grounded but the figure 1 (line 352) should be reformulated in “_ _ + and _ _ _ “ is not clear the (-) and it refers a figure 3 that appears much later (line 526).

Reviewer 3 Report

The manuscript sent to me for review, entitled "Sustainable mechanism of the Entrusted Transportation Management Mode on High-Speed ​​Rail and the impact of covid-19: A Case Study of the Beijing-Shanghai High-Speed ​​Rail" is generally good.

My comments mainly concern methodological issues:

  1. Hypotheses 1-3 (lines 404-409) are incorrectly formulated. These are conclusions, not hypotheses. The statements used in them are too obvious, they do not require inference.
    • The research shows that the authors investigated the cause-effect relationships, therefore the hypothesis should be cause-effect, have a suspecting element in it, i.e. be verified in the course of the study. For example, COVID-19 influence the sustainable development of HSR or ... (factors - you should enumerate) influence the sustainable operations of HSR passenger transport. It has to be in the present tense
  2. The formulas used in the manuscript are incorrect or incorrectly written. Rather, they are spreadsheet formulas. Nevertheless, authors should give them a record of agreement with mathematical or statistical standards. It is worth introducing variables and describing them under the formulas. In some of the formulas it will be necessary to enter the matrix notation. Where necessary, please use fractional notation, power notation, etc. You must also provide the source of the formulas (where they can be found in the literature). Characters like * / ^ are not valid here. Pay attention to all patterns, but especially formula 6, 19 and 24.
  3. In my opinion the sentence (in lines 89-91) "Taking scenarios with different severity of COVID-19 and analyze the impact of COVID-19 on sustainable development of Beijing-Shanghai HSR" is an aim of paper. You should point that: The research aim of paper is: .... (sentence above)
  4. Rephrase the sentences "Fiebel pointed", "Fumio pointed out", "Change pointed", "Wu analyzed" without using surnames.
  5. Data sources ("Beijing-Shang-467 hai HSR Co., Ltd. Annual Report 2019", "Beijing-Shanghai HSR Co., Ltd. Initial 468 Public Offering Prospectus") should be added to the reference.
  6. In table 7, you need to explain the empty fields and the sign "/" in the foot of the table.
  7. Move the heading of table 3 to the next page.
  8. Move the headline of figure 1 to the previous page.
  9. The line 605 annual should be lowercase.
  10. Rewrite the tasks in lines 664 and 704 so that they are incorporated into the following sentences.
  11. Adapt the list of references to the journal's requirements.
